# Dissection of Hyperspectral Reflectance to Estimate Photosynthetic Characteristics in Upland Cotton (*Gossypium hirsutum* L.) under Different Nitrogen Fertilizer Application Based on Machine Learning Algorithms

**DOI:** 10.3390/plants12030455

**Published:** 2023-01-19

**Authors:** Peng Han, Yaping Zhai, Wenhong Liu, Hairong Lin, Qiushuang An, Qi Zhang, Shugen Ding, Dawei Zhang, Zhenyuan Pan, Xinhui Nie

**Affiliations:** 1Key Laboratory of Oasis Ecology Agricultural of Xinjiang Production and Construction Corps, Agricultural College, Shihezi University, Shihezi 832003, China; 2Research Institute of Economic Crops, Xinjiang Academy of Agricultural Sciences, Urumqi 830091, China

**Keywords:** cotton, nitrogen monitoring, absorbed photosynthetically active radiation, net photosynthetic rate, remote sensing monitoring, machine learning algorithms models

## Abstract

Hyperspectral technology has enabled rapid and efficient nitrogen monitoring in crops. However, most approaches involve direct monitoring of nitrogen content or physiological and biochemical indicators directly related to nitrogen, which cannot reflect the overall plant nutritional status. Two important photosynthetic traits, the fraction of absorbed photosynthetically active radiation (FAPAR) and the net photosynthetic rate (Pn), were previously shown to respond positively to nitrogen changes. Here, Pn and FAPAR were used for correlation analysis with hyperspectral data to establish a relationship between nitrogen status and hyperspectral characteristics through photosynthetic traits. Using principal component and band autocorrelation analyses of the original spectral reflectance, two band positions (350–450 and 600–750 nm) sensitive to nitrogen changes were obtained. The performances of four machine learning algorithm models based on six forms of hyperspectral transformations showed that the light gradient boosting machine (LightGBM) model based on the hyperspectral first derivative could better invert the Pn of function–leaves in cotton, and the random forest (RF) model based on hyperspectral first derivative could better invert the FAPAR of the cotton canopy. These results provide advanced metrics for non–destructive tracking of cotton nitrogen status, which can be used to diagnose nitrogen nutrition and cotton growth status in large farms.

## 1. Introduction

Nitrogen is an important component of the structural and vital substances of crops, is an essential element for crop metabolism, and affects the growth state of crops [1,2,3]. Nitrogen deficiency leads to a decrease in leaf thickness [4], closure of leaf stomata [5], degradation of photosynthetic enzymes and thylakoids [6], decrease in chlorophyll content [7,8], and obstruction of light–energy capture, electron transfer, and carboxylation rate processes in leaves [9]. Simultaneously, the growth and development of nitrogen–deficient plants are hindered; plants are short and thin and the leaf area is reduced, which reduces the utilization efficiency of photosynthetically active radiation (PAR) in the crop canopy [10]. Excessive nitrogen application leads to the close arrangement of leaf mesophyll cells [11], which is not conducive to the entry of external CO_2_ into the cells, resulting in a decrease in the photosynthetic rate and even toxic effects on crops [12]. In addition, excessive nitrogen application can lead to excessive leaf numbers and dense canopies, which reduces canopy absorption and light energy utilization in crops [13,14]. The fraction of absorbed photosynthetically active radiation (FAPAR) and the net photosynthetic rate (Pn) are effective tools to assess the response of crops to nitrogen changes at the canopy and leaf scales, respectively [15,16,17,18,19,20,21]. It is of great significance to monitor the growth status of crops to determine their nitrogen nutrition status through changes in crop photosynthetic capacity. Traditional methods rely on in–field gas–exchange systems [22] or optical instrument measurement [23,24]. Such approaches can provide precise photosynthetic information, but are costly, time–consuming, and difficult to accomplish, especially in large cotton–growing areas. Therefore, pre–symptomatic or pre–visual detection of plant physiological changes is urgently required to avoid severe damage.

Remote sensing monitoring, as the current mainstream mode of detection, has the unique advantages of being rapid, non–destructive, and multidimensional, and is widely used in the agricultural production process [25,26,27,28]. The spectral characteristics of the plant canopy reflect various types of state information during plant growth [7,29,30]. Research has indicated that the spectral characteristics of plant canopy reflections are closely related to several biochemical and biophysical properties, such as pigment concentrations [31,32], plant vigor [33,34], water status [35,36], and nutritional status [37,38]. Different nitrogen states in cotton cause reflectance changes in multiple bands of the spectrum, such as 550–700 nm [39], 705–715 nm [40], and 1325–1575 nm [41]. Hyperspectral remote sensing plays an important role in plant nitrogen nutrition monitoring as a cutting–edge technology. This has been reported for many plant species, and the hyperspectral method has been used to provide rapid, non–destructive estimates of nitrogen status in most major crops, including rice [42,43], maize [44,45], wheat [26,46], and others [27,41,47]. Many studies have confirmed that the photosynthetic ability of crops, which is an important indicator of crop growth status, can be used to indirectly assess growth status [5,20,48]. At the same time, researchers have also constructed hyperspectral estimation models of the photosynthetic characteristics of different crops. Zhou et al. [36] used hyperspectral remote sensing to monitor the photosynthetic characteristics of citrus Pn under water stress. Jin et al. [49] used the hyperspectral vegetation index to evaluate the maximum carboxylation and electron transfer rates of alpine deciduous forests. There are some examples of monitoring plant photosynthetic characteristics using hyperspectral applications [7,50,51]. However, the current research on the nitrogen nutrition status of cotton primarily uses plant or leaf nitrogen content, dry matter, and photosynthetic pigments as direct monitoring indicators, and does not take into account changes in cotton physiology and morphology. Therefore, it is meaningful to study how to use remote sensing to monitor cotton photosynthetic parameters to assess the nitrogen nutrition status effectively and non–destructively.

The aim of this study was to construct a remote sensing estimation model for photosynthetic characteristic parameters and population photosynthetically active radiation in cotton leaves under different nitrogen levels. During the key growth period of cotton, the photosynthetic parameters of cotton leaves, population photosynthetically active radiation, and canopy spectral information were obtained. By mining spectral information, the optimal modeling method and hyperspectral variables were used to construct an estimation model of the photosynthetic characteristic parameters of cotton leaves and the population’s photosynthetically active radiation. This study provides a theoretical basis for efficient and non–destructive monitoring of the nitrogen nutrition status of cotton and provides scientific guidance for specifying scientific and reasonable fertilization programs.

## 2. Results

### 2.1. Data Distribution of Cotton Photosynthetic Characteristics Parameters

To obtain a dataset with variations, the spectral reflectance, Pn, and FAPAR of the cotton canopy for two cotton varieties were obtained at five growth stages under four nitrogen fertilizer applications across 2 years. A total of 240 data pairs were collected, with 30 pairs for each factor level. In all experiments, Pn and FAPAR were in the range of 20.51–37.91 µmol CO_2_·m^−2·^s^−1^ and 0.7949–0.9925, respectively (Figure 1A,B). The dataset for subsequent modeling was divided into training (n = 180) and validation (n = 60). The training and validation datasets were balanced to prevent bias in regression and metrics. Spearman’s correlation analysis showed that there was a significant positive correlation between Pn and FAPAR (Figure 1C,D).

### 2.2. Preprocessing and Dimensionality Reduction for Hyperspectral Data

Dimension reduction and preprocessing are important methods for ensuring the reliability and efficiency of a model. Principal component analysis showed that the first eight principal components contributed to 99% of the variability across the reflectance spectrum (Figure 2A). The top ten loading factors, as the main contributors to these principal components, were statistically analyzed. The results showed that the three regions of the hyperspectral characteristics of cotton affected by the nitrogen application level were 350–450 nm (accounting for 18.25% of the total bands), 650–700 nm (32.75%), and 1300–1350 nm (15.00%) (Figure 2B). The hyperspectral data of cotton at different stages of growth were then analyzed for band autocorrelation (Figure 2C and Appendix A). We selected 2000 combinations of bands with the lowest correlation; the frequency of each band was counted, and the hyperspectral bands with rich information were in the ranges of 350–450 nm (25.29%), 600–750 nm (39.94%), and 750–1000 nm (23.62%) (Figure 2D). Previous studies have shown that 350–450 nm and 600–650 nm are strong absorption bands of chlorophyll and carotenoids, and 700–750 nm is related to the content of chlorophyll (a + b) per unit area of plants [52]. Therefore, the high spectral reflectance of 350–450 and 600–750 nm was considered the best band range for model construction.

By comparing the basic soil fertility of the two–year experimental area, the total nitrogen content and organic matter content of the experimental area in 2019 were lower than those in 2020 (Appendix A). To avoid insignificant spectral differences between different treatments due to the high base fertility of the soil, the spectral reflectance data of the 2019 experiment (Test2019) were chosen for analysis. The 1350–1480 and 1780–1990 nm wavelengths were removed because the water vapor absorption peak has a large effect on the reflectivity of the cap. The variation trends of the canopy hyperspectral reflectance of the two cotton varieties in different growth periods were similar (Figure 3A and Appendix A), but there were significant differences in the canopy spectral reflectance in different wavelength ranges. There is a large difference in the reflectivity between the visible and near–infrared bands. Five preprocessing methods, namely, first–derivative reflectance (FD), second–derivative reflectivity (SD), wavelet transform reflectivity (WT), Savitzky–Golay smoothed reflectivity (SG), and natural logarithm reflectivity (NL), were applied to the original spectral reflectance (OR) data to build a regression model. Compared with the original spectral reflectance, the derivative–type spectra reduced the intra–class variance by removing noise from the original spectral data (Figure 3B,C and Appendix A). After preprocessing the spectral data using two smoothing methods, small variations in reflectance were eliminated across the wavelength range, making it more uniform (Figure 3D,E and Appendix A). With the natural logarithmic change, subtle differences in the original spectral curve became more pronounced (Figure 3F and Appendix A).

### 2.3. Machine Regression Models of Pn and FAPAR Based on Characteristic Bands

To simplify the model and reduce over–fitting, the reflectance wavelengths of 350–450 and 600–750 nm reflectance were used as characteristic bands to construct the model. Correlation estimation models of cotton leaf Pn and canopy FAPAR were constructed using four different machine learning algorithms: Support vector machine (SVM), random forest (RF), extreme gradient boosting (XGBoost), and light gradient boosting machine (LightGBM). Model performance was evaluated using the ratio of performance to deviation (RPD) values and robustness to multiple repetitions. The machine learning models indicated high performance and robustness when first–derivative reflectivity and second–derivative reflectivity were applied as hyperspectral data in the training and validation datasets. For Pn modeling, RF, XGBoost, and LightGBM models indicated high performance and robustness when FD was applied as hyperspectral data both in the training data and validation data, as most RPD values were greater than 1.4, representing an acceptable prediction level (Figure 4A and Figure 5A). For FAPAR modeling, RF and LightGBM models based on FD also demonstrated high performance and robustness in training and validation data, with RPD values above 2.0, representing an accurate prediction level (Figure 4B and Figure 5B). These results were supported by the coefficient of determination (R^2^), root mean square error (RMSE), and mean absolute percent error (MAPE), which are indicators of model performance (Appendix A).

## 3. Discussion

Nitrogen management is problematic in large–scale cotton cultivation. However, laboratory methods for measuring nitrogen content and photosynthetic characteristics remain complex experimental techniques that are not suitable for large–scale and real–time monitoring [22,23,24]. Therefore, novel techniques are required to efficiently assess nitrogen nutrient status and monitor crop photosynthetic physiology. The nitrogen nutrition status of plants is not only reflected in the changes in nitrogen content in plants, but also in changes in plant growth status, including changes in the internal physiology and external morphology of plants, which can be reflected in plants’ spectral signature [29,50,53]. Although using remote sensing to assess plant responses to environmental changes is a very active research topic, most studies to date have focused on nitrogen content or hormone content related to nitrogen stress [54,55]. At present, there are few studies on the correlation between cotton photosynthetic characteristics and hyperspectral data for estimating the nitrogen nutrition status of cotton. Therefore, the current study aimed to address these questions by using the difference in hyperspectral reflectance caused by different nitrogen application conditions, using a machine learning algorithm to accurately estimate cotton Pn and FAPAR, and implementing non–destructive estimation of cotton nitrogen nutrition status.

Hyperspectral techniques can obtain detailed spectral information from plants and can be used to closely monitor plant growth as well as physiological and biochemical properties [41,50,53,56]. Nevertheless, the use of full hyperspectral bands may cause information redundancy and interference [57,58]. In this study, principal component analysis and band autocorrelation based on the original spectral reflectance jointly screened the characteristic bands. Two highly correlated band regions (350–450 and 600–750 nm) were co–selected, located in the visible bands. Previous research has shown that the light depth in these wavelength ranges is involved in the photosynthetic process of plants [59,60], and the reflectance can reflect photomorphogenesis in and out of plants [61,62].

At the same time, it is necessary to use preprocessing methods to improve the level of hyperspectral modeling. Preprocessing of original spectral data was performed using the first derivative, second derivative, wavelet transform smoothing, Savitzky–Golay smoothing, and natural logarithm transformation to reduce the effect of background noise. These preprocessing methods, based on the full–band original spectral dataset, retained all the information of the original spectral data. Our research shows that the accuracy and stability of the prediction model were improved by using suitable preprocessing and transformation and that the machine learning model built using derivative–type spectral data was better than other preprocessing methods. Peng et al. believed that when the distance between the sensor and the collected crops was small, the low–order differential could effectively remove spectral background noise and improve the inversion ability of the model [63]. Xia et al. also reported that the application of derivative spectroscopy improved the accuracy of the model [64].

Based on a powerful and flexible algorithmic framework, machine learning has the potential to analyze hyperspectral reflectance data, thereby avoiding the waste of hyperspectral information in traditional research [30,65,66]. In this study, most FAPAR models were more robust than Pn models. The RPD values of the predicted data for most FAPAR models were greater than 1.4, which is an acceptable prediction level. This may be related to the collection form of hyperspectral data. Canopy–level hyperspectral information has a stronger predictive ability for FAPAR; its predictive ability for Pn requires improvement.

The machine learning methods LightGBM, RF, and XGBoost showed high performance and robustness for both Pn and FAPAR. The excellent performance of LightGBM may be attributed to its unique leaf growth strategy. RF is more robust for input variables and outliers. XGBoost adds a regular term to control the complexity of the model and reduce the possibility of overfitting. Therefore, machine learning is a powerful tool for estimating crop–related indicators using hyperspectral technology. In recent years, machine learning models have been increasingly used in agricultural remote sensing [41,65,66,67]. However, in previous studies, a single band or vegetation index was often used to remotely estimate crop growth indicators, resulting in a large waste of hyperspectral data and unstable model predictions [55,68,69]. In this study, we used a larger range of characteristic bands to predict the cotton nitrogen nutrient status. This will help improve the difficulty of monitoring large areas of cotton. This study contributes to the establishment of non–destructive monitoring management strategies and accurate modeling methods.

## 4. Materials and Methods

### 4.1. Site Characteristics

The experiments were conducted from May to September 2019 (Test2019) and 2021 (Test2021), respectively, in two experimental fields with different basal fertilities (Appendix A) at the field experiment station of the College of Agriculture, Shihezi University, Shihezi, Xinjiang Uygur Autonomous Region, China (43°26′–45°20′ N, 84°58′–86°24′ E). Shihezi is a hinterland city in Xinjiang with a typical temperate continental climate. The annual average temperature, annual average relative humidity, precipitation, and annual average frost–free period are approximately 7 °C, 55%, 180 mm, and 170 days, respectively. The rainfall and temperature during the cotton growing season and the time period of the experiment (from April to October) are shown in Appendix A.

### 4.2. Experimental Design

The experiment was conducted using a split–plot experimental design of randomized groups with three replicates, which included treatments with four amounts of nitrogen fertilizer as follows: (i) No nitrogen application (N0), without N fertilization application; (ii) less nitrogen application (N1), N fertilizer applied at 150 kg N ha^−1^; (iii) common nitrogen application (N2), N fertilizer applied at 300 kg N ha^−1^; and (iv) excessive nitrogen application (N3), nitrogen fertilizer applied at 450 kg N ha^−1^. The sub–area treatments with cotton of two different genotypes were as follows: (i) Xinluzao61 and (ii) Xinluzao72.

Except for nitrogen fertilizer, the other fertilizers, phosphate (P) and potassium (K), were the same for all treatments: 105 kg P_2_O_5_ ha^−1^ and 75 kg K_2_O ha^−1^. The application rates were recommended by local agricultural technicians and were sufficient to meet the requirements of cotton. The fertilizer types used in this study were urea (46.4% N), potassium dihydrogen phosphate (52% P_2_O_5_, 34% K_2_O), and potassium sulfate (52% K_2_O). All fertilizers were applied in multiple drops of water (Appendix A).

Two cotton varieties were selected (Xinluzao 61 and Xinluzao 72); both have excellent variety characteristics and are the major cultivars in Xinjiang (Appendix A). A plastic film mulch and drip irrigation were used in the experiments. Before sowing, the experimental plots were covered with 2.28 m–wide sheets of transparent plastic film. Drip irrigation lines were installed beneath the plastic film. The cultivars were sown through holes in a plastic film mulch. A wide–narrow row–spacing (66 + 10 cm) configuration was used in the experimental plots, with a planting density of 174,000 plants ha^−1^. Weeds, pests, and diseases were controlled using chemicals via local methods. No obvious weed, pest, or disease problems occurred during the cotton–growing season.

### 4.3. Photosynthetic Measurements

Following the standard method [22], cotton reached peak photosynthesis at 11:30–13:30 Beijing time on a day with sunny and stable weather. The net photosynthetic rate (Pn, µmol CO_2_·m^−2^s^−1^) of cotton key functional leaves was measured on clear and cloudless days during five reproductive periods using an LI–6400XT portable photosynthesis instrument (LI–COR Bioscience Inc., Lincoln, NE, USA). The data were collected on the 55th, 70th, 85th, 100th, and 115th days after cotton emergence, involving the full bud stage, full flowering stage, flower and boll stage, full boll stage, and early boll–opening stage of cotton. The effective radiation of the light source was PAR 1800 μmol·m^−2^s^−1^, the leaf chamber used was 2 × 3 cm, the gas flow rate was 500 mmol·s^−1^, and the CO_2_ injector was 400 μmol·mol^−1^. Ten plants from each treatment group were centrally selected for each measurement period.

PAR was synchronously sampled with the leaf net photosynthetic rate and measured in the time domain. PAR was measured using an LI–250A light meter with an LI–190SA quantum sensor (LI–COR Bioscience, Inc., Lincoln, NE, USA). According to previous methods [23,24], the measured solar radiation included incoming PAR above the canopy (PARt) and the three types of PAR transmitted through the canopy to the ground (PARs, PARn, and PARw). PARt was measured using a quantum sensor placed horizontally 0.5 m above the cotton canopy surface pointing toward the sky; PARs were measured with line quantum sensors placed approximately 5 cm above the ground, pointing upward; PARw and PARn were measured with line quantum sensors placed approximately 5 cm above the ground in wide and narrow rows of cotton, respectively, pointing upward. Each type of PAR was obtained using the mean value of the three different measurement directions. The average values of cotton PARs, PARw, and PARn were taken as the cotton–bottom PAR (PARb). The fraction of absorbed photosynthetically active radiation (FAPAR) was determined using Equation (1).
(1)$$\mathrm{FAPAR}=\frac{\left({\mathrm{PAR}}_{t}-{\mathrm{PAR}}_{b}\right)}{{\mathrm{PAR}}_{t}}$$

### 4.4. Hyperspectral Measurement Processing

For synchronization with the measurement of photosynthesis, the spectral radiance of the cotton canopy was measured with a full–range hyperspectral SR–3500 portable ground object spectrometer (Spectral Evolution, Lawrence, KS, USA) under natural light on a sunny day in the field. The instrument collected data in the 350–2500 nm spectral range, with a resampled spectral resolution of 3 nm before 1000 nm, 8 nm between 1000 and 1500 nm, and 6 nm after 1500 nm.

For cotton plant canopies where photosynthesis was measured, the spectral reflectance was measured 50 cm above the cotton canopy using a hyperspectral one–handed remote–measuring device. The acquired reflectance spectra could be affected by several environmental factors. Whiteboard calibration was used to eliminate or minimize these side effects [70]. Before each measurement, the instrument was optimized once, and a standard whiteboard calibration was carried out. The reflectance of the cotton canopy was computed as the ratio of leaf radiances relative to the radiance from the white reference panel. More than five measurements at each location and averaged to produce a spectral reflectance. Equation (2) was used for cotton canopy reflectance calculation.
(2)$$R_{ci}=\frac{R_{mi}}{R_{wi}}$$
where $R_{ci}$ is the corrected reflectance of the cotton canopy, $R_{wi}$ is the reflectance of the whiteboard, and $R_{mi}$ is the measured reflectance of the cotton canopy.

Owing to background noise, the signal–to–noise ratio of the original spectral reflectance was low. Smoothing, derivative processing, and mathematical transformation are considered to be important for improving the signal–to–noise ratio [71,72]. To be more conducive to the recognition of original spectral data, five types of preprocessing methods, including first derivative, second derivative, wavelet transform smoothing, Savitzky–Golay smoothing, and natural logarithm transformation, were adopted to eliminate data noise and highlight the change law of reflectance with wavelength.

### 4.5. Filtering Characteristic Bands

Using the splice correction function that comes with the instrument, reflectance data at 1–nm steps were obtained across the entire wavelength domain from 400 to 2500 nm. Hyperspectral data contain information on many bands and there is large data redundancy and collinearity, which can affect model building [57,58]. Thus, it is necessary to reduce data redundancy by reducing data dimensionality. To be more conducive to the screening of the hyperspectral feature band range, the dimensionality reduction methods of principal component analysis and band autocorrelation were used to select the feature band.

Principal component analysis can analyze the main factors that cause differences in a large number of variables, and then achieve the effect of data dimensionality reduction [73,74]. By screening the bands corresponding to the top ten weight coefficients of each principal component that provided more than 99% of the spectral variation interpretation, the optimal band for the spectral information of the cotton canopy was selected.

Band autocorrelation can reduce spectral content redundancy through the correlation relationship between the two bands [75]. The smaller the correlation coefficient, the lower the information redundancy between the two bands. Using the spectral reflectance in the 350–2500 nm band, the r^2^ correlation coefficient was calculated using a pairwise combination, and all the combinations constituted a 2151 × 2151 r^2^ matrix. By selecting the band combinations corresponding to 2000 smaller values in the r^2^ matrix of different growth periods, the occurrences of different bands were counted, and the band range with rich hyperspectral information was determined according to the band distribution.

However, various methods have been devised to select the range of characteristic bands, each with its own advantages and limitations. Therefore, in this study, we consider the common part of the characteristic bands screened by the two methods as the characteristic band range.

### 4.6. Model Construction

The process of generating regression models is described in a traditional manner [76], with slight modifications. For modeling, the data from Test 2019 and Test 2021 were divided into two groups by stratified sampling (Figure 6): The training dataset (60%) and the validation dataset (40%).

Based on the feature bands screened by band autocorrelation analysis and principal component analysis, correlation estimation models of cotton leaf Pn and canopy FAPAR were constructed using four different machine learning algorithms, including SVM, RF, XGBoost, and LightGBM with spectral OR, FD, SD, WT, SG, and NL as independent variables.

SVM can better minimize the structural risk and select the optimal solution between the accuracy of the given data and the complexity of the function to obtain the best model promotion ability. The SVM model has the advantage of solving the problems of the small sample size and nonlinear, high–dimensional data [77].

The decision tree is a common machine learning algorithm that is easy to understand and interpret. RF builds a bagging ensemble with a decision tree as the base learner and further introduces random feature selection in the training process of the decision tree, which has great advantages in processing high–dimensional data [77]. XGBoost is a large–scale parallel boosting tree tool that has the advantages of rapid speed, good effect, and the ability to handle large–scale data. Its regularization, parallel structure, and high flexibility have introduced the ability to improve the prediction model [78]. LigthGBM is a new member of the boosting set model that adopts the strategy of leaf–wise tree growth to construct decision trees and features ultrafast efficiency in coping with large datasets [79]. LightGBM has shown potential as an efficient model method for various agricultural targets using reflectance data such as seed germination [80], lignin content [81], variety classification [82], and crop breeding [83].

### 4.7. Data Analysis and Charting

The preprocessing of the five types of hyperspectral data was performed using Origin 2022b (OriginLab, Northampton, MA, USA) and Microsoft Excel 2019 (Microsoft Corporation, Redmond, WA, USA). The SG smoothing window size was set to 5 [84]. We set the wavelet type to DB6, the expansion mode to periodic, and the truncation rate to 20% in the WT smoothing [85].

Four machine learning algorithms were performed using the “mlr3” package (V. 0.13.3) in R 4.1.3 [86]. This package integrates various machine learning algorithm packages to achieve a unified and neat machine learning process–oriented operation. To optimize these machine learning algorithms, we used ten–fold cross–validation and a random search to tune the hyperparameters. Information about the hyperparameters of these machine learning algorithms is presented in Appendix A. All figures were created using the R package “ggplot2” V. 3.3.5 [87] and the extension packages. Statistical analyses were conducted using the R statistical programming language V. 4.1.3.

### 4.8. Model Testing and Evaluation

The prediction metrics used to evaluate the abovementioned algorithms were R^2^, RMSE, and MAPE, which were calculated from Equations (3)–(5).
(3)$$\mathrm{RMSE}=\sqrt{\frac{\sum_{i=1}^{n}{\left(y_{i}-{\hat{y}}_{i}\right)}^{2}}{n}}$$
(4)$$R^{2}=1-\frac{\sum_{i=1}^{n}\sum {\left(y_{i}-{\hat{y}}_{i}\right)}^{2}}{\sum_{i=1}^{n}\sum {\left(y_{i}-\bar{y}\right)}^{2}}$$
(5)$$\mathrm{MAPE}=\frac{1}{n}\sum_{i=1}^{n}\left|\frac{\hat{y_{i}}-y_{i}}{y_{i}}\right|\times 100\%$$
where $\hat{y_{i}}$ is the predicted Pn or FAPAR value, $y_{i}$ is the measured Pn or FAPAR value, $\bar{y}$ is the mean value of the measured Pn or FAPAR, and n is the number of validations.

In the field of spectroscopy, the RPD has been used as the standard method to report the quality of a model [37,88,89,90,91]. The quality of the prediction model was interpreted according to three classes of RPD as follows: RPD > 2 indicates accurate model predictions, RPD of 1.4–2 indicates reasonably acceptable predictions, and RPD < 1.4 indicates poor model predictions. RPD was conducted using “yardstick” (R package).

## 5. Conclusions

In this study, correlation estimation models between hyperspectral data and photosynthetic characteristic parameters (Pn and FAPAR) were established using machine learning algorithms under different nitrogen conditions. The results showed that hyperspectral reflectance data and machine learning methods have excellent potential for estimating leaf Pn and canopy FAPAR in cotton. Among them, the LightGBM model (selected super–parameters were the shrinkage rate: 0.053, number of boosting iterations: 479, maximum number of leaves in one tree: 22, limit the maximum depth for tree model: 5, and minimal number of data in one leaf: 18), based on the hyperspectral first derivative, can better invert the Pn of cotton leaves. The RF model (selected super–parameters were the number of trees: 87; number of variables used to split the nodes: 68; minimum number of unique cases in a terminal node: 3; and maximum depth at which a tree should be grown: 37), based on the hyperspectral first derivative, can better invert the FAPAR of the cotton canopy. These results provide advanced metrics for non–destructive tracking of crop nitrogen status. In the future, these techniques can be used to diagnose nitrogen nutrition and the growth status of crops on large farms in real time.

## Figures and Tables

**Figure 1 plants-12-00455-f001:**
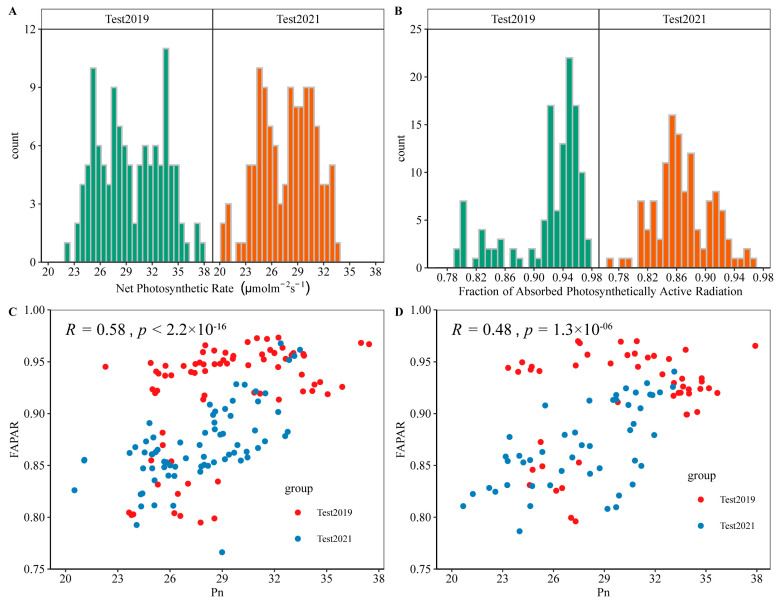
Data distributions of net photosynthetic rate (Pn) and fraction of absorbed photosynthetically active radiation (FAPAR) in cotton. (**A**) Histograms of Pn distribution in all experiments. (**B**) Histograms of FAPAR distribution in all experiments. (**C**) Correlation plots of Pn and FAPAR contents in training dataset. (**D**) Correlation plots of Pn and FAPAR contents in validation datasets. Statistical results for correlation are shown.

**Figure 2 plants-12-00455-f002:**
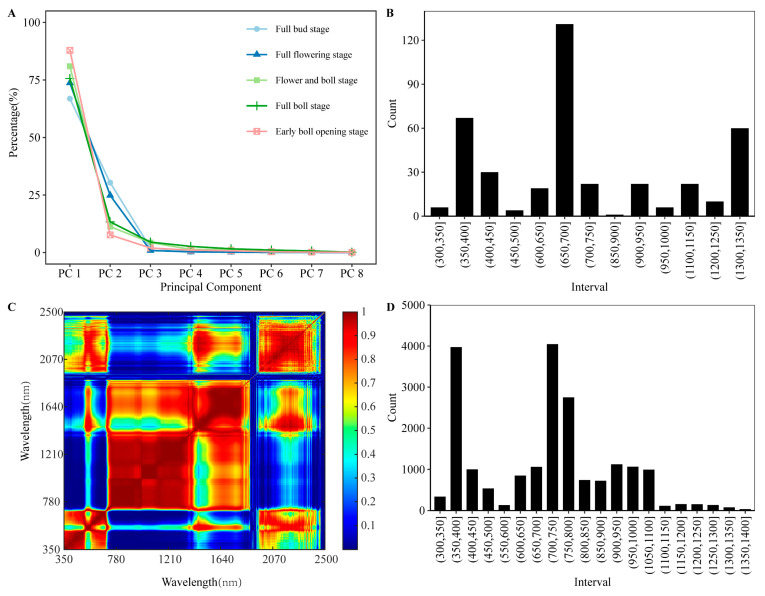
Dimensionality reduction performed using principal component analysis and band autocorrelation analysis. (**A**) Contribution of each principal component to total variation in cotton hyperspectral. (**B**) Frequency distribution of bands selected by the first eight principal components in each band range. (**C**) Decision coefficient of autocorrelation of spectral reflectance in different bands of cotton canopy at full bud stage. (**D**) Distribution of autocorrelation selected bands in each band range.

**Figure 3 plants-12-00455-f003:**
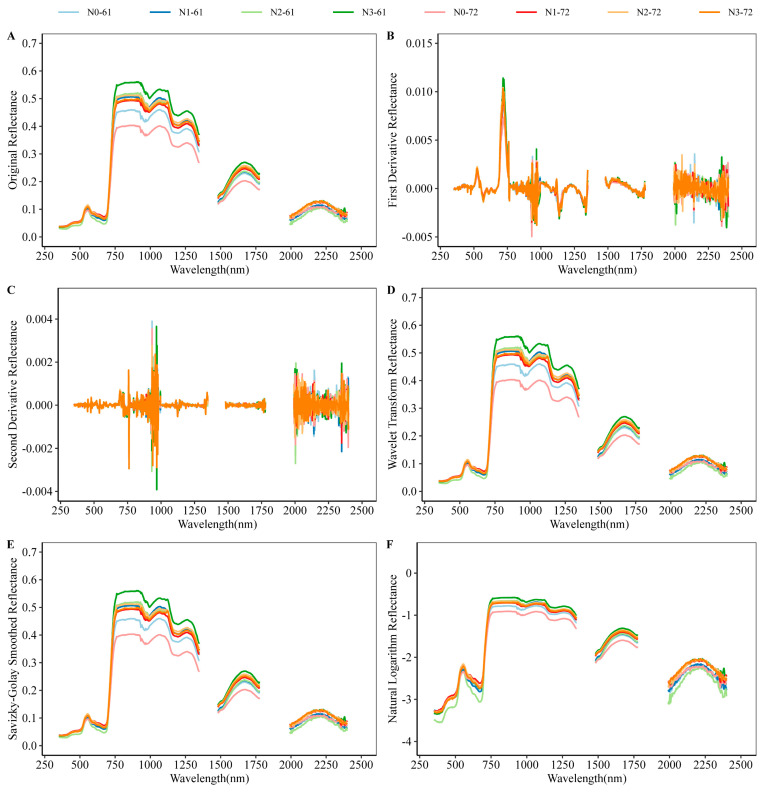
Original and preprocessed spectral data of two cotton varieties under different nitrogen treatments at full bud stage in Test2019. (**A**) Spectral original reflectance; (**B**) first–derivative reflectivity; (**C**) second–derivative reflectivity; (**D**) wavelet transform reflectivity; (**E**) Savitzky–Golay smoothed reflectivity; (**F**) natural logarithm reflectivity.

**Figure 4 plants-12-00455-f004:**
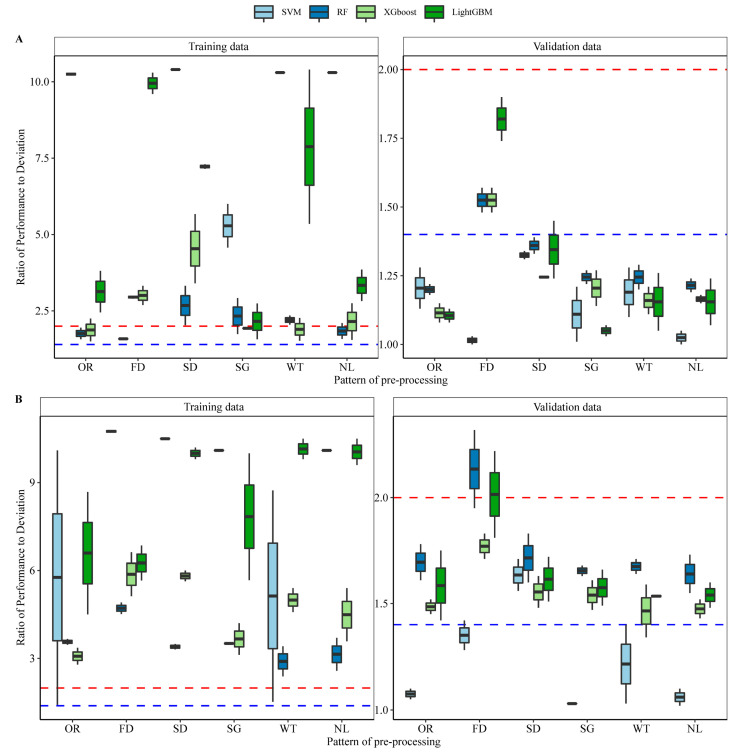
Model performance and robustness for each preprocessing of reflectance. (**A**) Machine learning models for Pn. (**B**) machine learning models for the fraction of absorbed photosynthetically active radiation (FAPAR). The ratio of performance to deviation (RPD) was applied to evaluate the accuracy of each model. Figures are plots of the RPD values in each repeat. Blue and red lines indicate RPD values of 1.4 and 2.0, respectively, as accuracy thresholds.

**Figure 5 plants-12-00455-f005:**
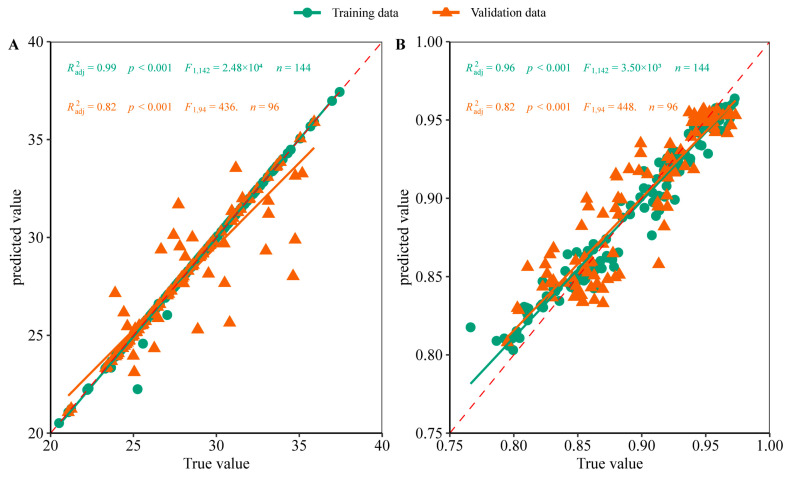
Measured vs. predicted values after applying machine learning models to predict net photosynthetic rate (Pn) and fraction of absorbed photosynthetically active radiation (FAPAR). (**A**) Pn–light gradient boosting machine (LightGBM); (**B**) FAPAR–random forest (RF). The red dotted line is the 1:1 line.

**Figure 6 plants-12-00455-f006:**
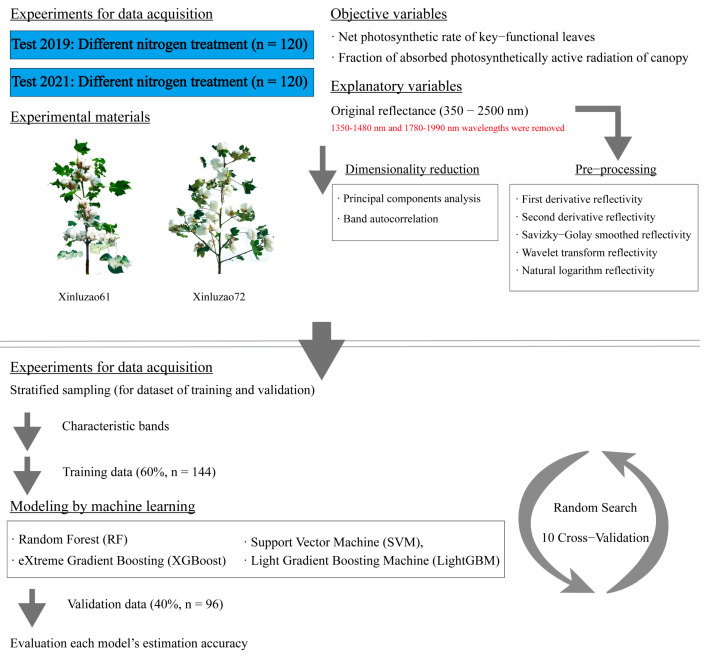
Experiment and modeling designs in this study.

## Data Availability

The raw data supporting the conclusions of this article will be made available upon request to the corresponding authors.

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
