# Peer review of "Dissection of Hyperspectral Reflectance to Estimate Photosynthetic Characteristics in Upland Cotton (Gossypium hirsutum L.) under Different Nitrogen Fertilizer Application Based on Machine Learning Algorithms"

_plants, 2023, doi:10.3390/plants12030455_

Round 1

Reviewer 1 Report

Dear colleagues,

I believe your manuscript may be useful and interesting, however, I am sure, there are some serious problems.

 (1) The title:

— "Cotton" is a name for fibers, you work with the cotton plants. Please, also add the scientific name for the species.

— What does "different nitrogen" mean? Nitrogen is the chemical element with several isotopes... Do you mean different isotopes?

 (2) In the beginning of the manuscript, you try to discuss the problems with nitrogen availability in agricultural systems and some problems associated with monitoring of this element. But the main goal of the text is "to dissect the relationship between hyperspectral reflectance and photosynthetic characteristics in cotton under different nitrogen based on machine learning algorithms." In the main text, you mainly discuss relationships between the hyperspectral reflectance and photosynthetic characteristics (I am sure that these correlations are evident); but not between the hyperspectral reflectance and nitrogen content. My guess is that nitrogen content is important for plants, but there are no direct associations between availability of this element and intensity of photosynthesis. I mean that is not good idea to use the hyperspectral reflectance to monitor nitrogen content. Actually  to monitor nitrogen content someone could try to use modern equipments allowing to estimate ATP content in the plants.

 (3) line 99 — Generally speaking, the Pearson correlation may be used for normally distributed data... Did you check normality?

 (4) lines 116–117 — Is there any biological meaning of the band 1300–1350 nm?

 (5) lines 262–263 — Humans commonly use the standard time, but living plants and animals do not know that, they use a real local time... For instance, in Shihezi, the real midday/noon is at 14:09! What's about the local peak of photosynthesis?

 (6) Please, check all situations when en-dashes should be used (e.g., for ranges, between years etc.).

 Besides, there are a lot of problems with English constructions and words, e.g.,

 lines 14–15 — nitrogen monitoring technology of crops

line 20 — hyperspectral ... (what?)

line 22 — spectral original reflectance > original spectral reflectance

lines 82–83 — cotton under different nitrogen based on machine learning algorithms

line 161 — batter > better...

 Please, check your text very carefully.

Reviewer 2 Report

A machine learning-based approach for understanding the different nitrogen conditions is offered in the proposal titled "Dissection of hyperspectral reflectance to estimate photosynthetic characteristics in cotton under different nitrogen conditions." The performance was assessed using four different machine-learning algorithms. 

The paper is well-written and the method is described very efficiently. There is a very detailed description of the overall experiment and dataset. There has been a significant improvement in the newer version of the paper compared to the original submission. 

Before this article can be further processed, there are a few questions that need to be addressed:

The Dimensionality reduction process must take into account biases present in the data, if any exist. 

Using traditional machine learning algorithms, you have achieved a very practical result. Do you have any experience with using state-of-the-art graph neural networks (GNNs) in the proposed experiment? For multidimensional data, GNNs proved to be very effective. 

Author Response

Dear reviewer,

Thank you for your affirmation of our manuscript. Please find below a detailed point-by-point response to all comments (reviewers’ comments in black, our replies in blue).

Question 1:

The Dimensionality reduction process must take into account biases present in the data, if any exist. 

Response: Thank you very much for your suggestion. In order to ensure that the variables involved in model construction can be explained and applied, we have chosen the method of feature screening for dimensionality reduction. In this way, the complete data characteristics can be preserved, and the bias in the original data can be avoided.

Question 2:

Using traditional machine learning algorithms, you have achieved a very practical result. Do you have any experience with using state-of-the-art graph neural networks (GNNs) in the proposed experiment? For multidimensional data, GNNs proved to be very effective.

Response: Thank you very much for your suggestion. Sorry, we do not have similar experience. GNNs are widely used in various applications and domains due to their effectiveness in modeling complex data structures, strong scalability, and high availability of methods. In future research, in order to better monitor the growth status of crops throughout the growth period, our experimental design includes more factors, such as more varieties, moisture in different gradients, different kinds of nutrients in several gradients, and so on. At that time, we will select deep learning methods including graph neural networks (GNNs) to build more accurate crop growth models.

Reviewer 3 Report

The authors of the manuscript titled "Dissection of hyperspectral reflectance to estimate photosynthetic characteristics in cotton under different nitrogen based on machine learning algorithms" present the results of a regression analysis comparing four machine learning algorithms. The manuscript would greatly benefit from proofreading and English editing, as some parts are difficult to understand, due to bad English. This includes the title. Furthermore, there are several issues with the manuscript which have to be resolved, before I could recommend it for publishing. All plots in the manuscript should be in vector graphics format, not raster. If the authors wish to persist with raster images, then they should be of good enough resolution. Currently, they are not and are sometines hard to read (example in Figure 2A). Also, I would recommend using the viridis colour scheme, as it is designed for colourblind readers. The Discussion is weak, I would expect more about the differences between pre-processing methods, why are second derivatives best in your case, a comparison between PCA and PLS or other dimensionality reduction methods. The general problem of overfitting in Decision tree-based algorithms and how they try to overcome it.

Detailed comments:
Lines 55-56: Why is it urgent? How much crop losses are attributed to too much or too little N? Is nitrogen heterogeneously distributed in fields? If so, would targeted fertilization be feasible?  

Line 75: You mention "many examples", but cite only 2. That's not many.

Lines 78-80: Which few studies? By using the keywords "cotton, nitrogen, hyperspectral, chlorophyll" Google Scholar finds 2930 hits since 2018. More than just a few of these deal with the same issues as your manuscript.

Lines 81-89: What were your hypotheses?

Line 98: How were the training and testing sets balanced? Across years or varieties? Both?

Line 99: The data in Figure 1 clearly shows a bimodal distribution (Fig. 1A), and in Fig. 1B only data for 2021 resembles a normal distribution. In Fig. 1C and 1D a clear distinction between years 2019 and 2021 is visible. Pearson's correlation coefficient requires a normal distribution of data. Did you normalize your data prior to analysis? Did you check for normality? This is also a requirement for regression analysis, even with machine learning. Why is there such a marked difference between the two years? Also, why not consecutive years, why is 2020 missing? Did you check the data for any outliers?

Line 112: PCA components don't contribute to variability. Individual bands contribute to PCA componenents, and the components explain a certain amount of variability in the data. Also, why 8 PCA components? According to Fig. 2A the first two to three explain more than 90% of the variability. Using too many components will lead to overfitting. Researchers have proven that even severe dimensionality reduction to just one or two components is enough for accurate machine learning models.  

Line 113: Why the top ten? How did you measure their contribution to PCA components? With correlations?

Lines 116-117: What are the percentages in the parentheses?

Lines 120-121: That's just chlorophyll a and b absorbance spectra, modified by accesory pigments. Nothing new.

Line 125: Figure 2A what is the "value" on the y-axis? Fig. 2C what are the labels of both axes? Why aren't tick marks on X and Y in the same positions? Why is correlation measured only between 0 and 1? Where are negative correlations?

Line 129: What ratio? In the text you stated that these are counts, not ratios.

Line 133: Why 2019 and not 2021? Were the regression models developed only on data from 2019 or both years?

Lines 136-137: Yes, because they are plants.

Lines 143-144: Figure 3 doesn't show that. This sentence would better fit in the discussion.

Line 146: What are the different colours in Figure 3? Don't just use ggplot2 default settings. Since this is data from only 2019, you could colour code the spectra to variety or treatment. Instead of individual lines you could also plot the mean/median spectra and 95% confidence limits as a ribbon. Why didn't you normalize the spectra, for example using area, mean or maximum normalization? This would minimize illumination effects.

Line 152: How does dimensionality reduction minimize overfitting? Also, later you state that your algorithms overfit the training data, so dimensionality reduction failed in this regard.

Line 156: What are FD and SD? I know they are explained later in methods, but as this manuscript is structured, this is the first time you mention these, so explain the abbreviations.

Line 163: This figure is hard to read, increase the size. What are the abbreviations on the right hand side? Why doesn't SVM have any error bars in the training dataset plots? Why does SVM perform better for Pn and LightGBM for FAPAR? Are the scales on the X-axis correct? For example, RMSE for training in A is between 0.0 and 2.0, while for validation it's between 1.0 and 3.0. Could the very evident ovefit in Pn have something to do with your bimodal distribution and non-normalized data? Error bars in the training set are larger than in testing. Why? Also, how did you measure robustness? The Kappa statistic would show you how well the different models from cross-validation overlap.

Line 168: Include basic regression statistics in the plot, such as R-squared for both sets. Fig. 5A shows a perfect overlap between actual and predicted values in the training set. Again, overfit. Are these regressions statistically significant? Minimum requirements for reporting regression analysis are R-squared, F-scores, N, degrees of freedom, p-values.

Line 183-185: Are you sure there are only few such studies? Same as for Lines 78-80.

Line 194: Again, this is just chlorophyll. Hardly a new finding.

Line 199: The method is called Savitzky-Golay. The same error occurs throughout the manuscript.

Line 200: What do you consider to be background noise when using a spectrometer? The distance between the sensor and leaf is measured in milimeters, so which background do you mean?

Line 201: How did you optimize the pre-processing methods, based on "full-band original spectral data"?

Line 202: Are you talking just about pre-processing methods, or dimensionality reduction? The paragraph starts with dimensionality reduction, so it isn't clear what you're talking about.

Lines 202-206: This isn't a new finding. It's well-known and any spectral analysis handbook will tell you to use pre-processing methods.

Lines 213-214: Did you mention overfitting in the results? Why not? What measures did you take to reduce overfit? Just because overfitting is a common problem doesn't mean you can just shrug it off. The models you developed are not reliable, due to overfitting. These models should not be used for assessing Pn and FAPAR in cotton, as they will lead to false results.

Lines 219-221: This sentence would fit better in the introduction.

Lines 224-226: This is the end of your discussion. Focus on your results, not others. Also, given the overfit problem in your results, did you really show that SVM and LightGBM are suitable for Pn and FAPAR regressions?

Lines 287-308: Provide the equation for calculating reflectance. Did you include the black (sensor background noise) signal in your calculation?

Lines 299-301: How does calculating reflectance, as a ratio between measured and incident radiance, reduce effects of reflectance off surrounding objects?

Lines 302-308: Why didn't you also use Multiplicative scatter corrections, standard normal variates, detrending, baseline corrections. Figure 3 shows that baseline corrections are needed, especially in Fig. 3F. There are several standard methods in spectroscopy to achieve similar results to spectral derivatives. Why didn't you test them as well?

Lines 307-308: What is the "distribution law of reflectance with wavelength"?

Lines 310-311: So you upsampled your original spectral data to a uniform bandwidth of 1 nm? Why?

Lines 310-317: Partial least squares regression is specifically designed to deal with data with high colinearity. And with spectral data it generally performs better than PCA. It is also a dimensionality reduction method. Why did you choose only PCA? Also, both PLS and PCA can be used directly for regression analysis. Their results can be similar or even better than machine learning methods.

Lines 314-317: This sentence is unclear. The manuscript really needs English editing and proofreading.

Lines 328-331: Sentense is unclear. How did you utilize this information? Did you construct models using only these bands? Were these bands used for PCA?

Lines 349-350: While a single decision tree is easily interpretable, random forests with hundreds if not thousands of trees are not. And as you mention in Line 351, RF also acts as a dimensionlity reduction method, since it randomly selects features. Using the Gini index you can then determine which features were selected more often and are therefore more important for your problem.

Line 359: Figure 6 shows that dimensionality reduction was performed prior to pre-processing. Is this correct?

Line 361-365: Why did you use these settings? A simmetrical window of 5 for SG can increase overfitting. Also, were spectra smoothed prior to calculating derivatives? Did you test different settings or just used default values?

Line 369: By using repeated cross-validation you would get more reliable results. Also, when reporting on machine learning results, include final values of tuned hyperparameters.

Round 2

Reviewer 1 Report

Dear authors,

I have some proposals to improve your manuscript:

line 2 - in cotton > in the upland cotton

lines 73, 77 and so on Zhou et al.... > Zhou et al. [37]...

line 118 65 > 650

lines 22, 117-119 and so on - your results show that 3 bands may be especially important: 350-450, 650-700 and 1300-1350 nm - please, try to comment/explain this pattern from the biological/ecological point of view. Why these bands may be important. Generally speaking, the first two bands more or less correspond two main bands of light absorbance during photosynthesis with chlorophyll a+b, but what's about 1300-1350 nm?

lines 145, 157 and so on (incl. Fig. 3) - Savizkg > Savitzky

Please, check all situations when en-dashes should be used (e.g., betweeb page numbers, between years etc.).

Author Response

Dear reviewer,

We thank the reviewers for their careful reading of the manuscript and their constructive remarks. We have taken the comments on board to improve and clarify the manuscript. Please find below a detailed point-by-point response to all comments (reviewers’ comments in black, our replies in blue).

Question 1:

  • line 2 - in cotton > in the upland cotton
  • lines 73, 77 and so on Zhou et al.... > Zhou et al. [37]...
  • line 118 65 > 650
  • lines 145, 157 and so on (incl. Fig. 3) - Savizkg > Savitzky
  • check all situations when en-dashes should be used (e.g., betweeb page numbers, between years etc.)

Response: Thank you very much for your suggestion. We have checked and corrected the mistakes through the text according your suggestion. Specially, the use of en-dashes has been carefully modified

Question 2:

lines 22, 117-119 and so on - your results show that 3 bands may be especially important: 350-450, 650-700 and 1300-1350 nm - please, try to comment/explain this pattern from the biological/ecological point of view. Why these bands may be important. Generally speaking, the first two bands more or less correspond two main bands of light absorbance during photosynthesis with chlorophyll a+b, but what's about 1300-1350 nm?

Response: Thank you for the questions. There were only two important wavelength ranges obtained in this paper, 350–450 and 600–750 nm. We are sorry for making you misunderstand. And, we have added the explanation of the patterns from the biological point of view in the manuscript: “previous studies have been shown that 350-450nm, 600-650nm are strong absorption bands of chlorophyll and carotenoids, and 700-750nm is related to the content of chlorophyll (a+b) per unit area of plants [53].” in lines 123-125 of page 4. At the same time, we delete “main” on line 117.